# The impact of signal variability on COVID-19 epidemic growth rate estimation from wastewater surveillance data

Ewan Colman[1], Rowland Kao[2]*

**1** Bristol Medical School, University of Bristol, Oakfield Grove, Bristol, United Kingdom, **2** Royal (Dick) School of Veterinary Studies and the Roslin Institute, University of Edinburgh, Easter Bush, Midlothian, United Kingdom

* rowland.kao@ed.ac.uk

## Abstract

Testing samples of wastewater for markers of infectious disease became a widespread method of surveillance during the COVID-19 pandemic. While these data generally correlate well with other indicators of national prevalence, samples that cover localised regions tend to be highly variable over short time scales. Here we introduce a procedure for estimating the real-time growth rate of pathogen prevalence using time series data from wastewater sampling. The number of copies of a target gene found in a sample is modelled as time-dependent random variable whose distribution is estimated using maximum likelihood. The output depends on a hyperparameter that controls the sensitivity to variability in the underlying data. We apply this procedure to data reporting the number of copies of the N1 gene of SARS-CoV-2 collected at water treatment works across Scotland between February 2021 and February 2023. The real-time growth rate of the SARS-CoV-2 prevalence is estimated at all 121 wastewater sampling sites covering a diverse range of locations and population sizes. We find that the sensitivity of the fitting procedure to natural variability determines its reliability in detecting the early stages of an epidemic wave. Applying the same procedure to hospital admissions data, we find that changes in the growth rate are detected an average of 2 days earlier in wastewater than in hospital admissions. In conclusion, this paper provides a robust method to generate reliable estimates of epidemic growth from highly variable data. Applying this method to samples collected at wastewater treatment works provides highly responsive situational awareness to inform public health.

## Introduction

Infectious disease surveillance is a fundamental pillar of public health management. Epidemiological data collected with regularity across multiple locations provides an awareness of the progress of a disease as it spreads through a population. Robust surveillance systems to support this will prevent adverse outcomes through early action, without undue risk of raising false alarms. An emerging technology that promises improve surveillance capabilities pathogen detection in wastewater [1]. This has been shown to provide early detection of

**Data availability statement:** RNA count from wastewater data are available from the Scottish Environmental Protection Agency, https://informatics.sepa.org.uk/RNAmonitoring/ Hospital admissions data in de-identified format

are available from https://doi.org/10.6084/m9.figshare.28512479.v1 Treatment site catchment data are available in the code repository, https://github.com/EwanColman/Estimating-epidemic-growth-rate-from-wastwater-data

**Funding:** EC was supported by Wellcome Trust (grant no. 209818/Z/17/Z) and the National COVID-19 Wastewater Epidemiology Surveillance Programme. RK received no specific funding for this work. The funders had no role in study design, data collection and analysis, decision to publish, or preparation of the manuscript. There was no additional external funding received for this study.

**Competing interests:** The authors have declared that no competing interests exist.

epidemic trends compared to other surveillance sources [2–6]. The technology has potential to detect a range of human diseases [7–13] and reach otherwise overlooked communities [14,15].

A central question for policy makers is whether incidence of a disease is increasing or decreasing. Since an upward trend in new infections may be a reason to introduce restrictions or adjust provisions to hospitals, it is important to have reliable methods for determining epidemic trajectories in real time from the available data [16–18]. For wastewater based epidemiology using RT-qPCR this is particularly challenging as there are factors beyond changes in the underlying population prevalence that affect the observed signal - an observed increase in the gene copy count between samples taken on different days does not necessarily indicate an increase in the number of infected people [19,20].

To mitigate the effects of natural variability, smoothing can be applied over time and space [18,21–24]. This approach can, however, have the unintended consequence of distorting some important features of the data, such as inflection points where there is a sudden change in the epidemic trajectory, or a sharp increase in one location that subsequently spreads to the wider region. We propose that an approach based on fitting an epidemiologically realistic model is more appropriate for these data as we would like the outcome to represent the underlying epidemic trajectory as accurately as possible. The question is how can we filter out the high levels of variability observed at the local scale, while retaining the ability to quickly detect changes in epidemic trajectory?

Within the existing literature, there are no studies that address the problem of inherent signal variability in designing a wastewater-based early warning system. This paper introduces a method to interpret the highly variable wastewater sample values and applies it to data from the COVID-19 pandemic in Scotland. We start by discussing the various causes of variability affecting data collected from wastewater samples and describe the distribution of of values we expect to observe at a given prevalence. We then introduce a function to measure the likelihood of any given epidemic trajectory in relation to the observed wastewater data and describe a procedure to find the best fitting function from a class of exponential models. We compare results across different sampling locations, optimise the process for early detection of epidemic waves, and finally compare results to hospital admissions data.

## Materials and methods

### Data

From February 2021 to the time of writing (February 2023), samples from Waste Water Treatment Plants (WWTP) were regularly taken by the Scottish Environment Protection Agency (SEPA) to detect fragments of SARS-Cov-2 virus RNA [25,26]. Samples from sewerage influent were taken at 122 sampling locations across Scotland approximately two to three days each week depending on the location [27]. Combining the catchment areas of all 122 WWTPs covers the households of approximately 4500000 people, 82% of the population of Scotland. The ten most populated catchments cover 47%.

Each Sample was collected using a refrigerated autosampler that obtained a fixed volume of influent every hour over a 24-h period (08:00 to 08:00). Composite 24-h samples were mixed and concentrated before viral RNA was extracted using commercial kits. SARS-CoV-2 N1 gene average concentrations (gene copies/l) are obtained using RT-qPCR.

WWTP data were provided by SEPA via a publicly available portal [28]. For each sample taken the data provides the date, the WWTP, and the N1 reported value obtained from RT-qPCR. Additional data for fluid flow for a limited number of WWTPs were provided though restricted access from SEPA. Since inflow to each WWTP comes from households and other

premises connected to the sewerage network, the recorded values relate to SARS-CoV-2 infections in a specific geographical catchment area. Scottish water provided shapefiles giving the border of the region corresponding to each WWTP.

Restricted access to hospital admission data was provided by Public Heath Scotland. The hospital admission database includes information on all individuals admitted to hospital in Scotland including primary reason for admission, other contributing conditions, date of admission, and the place of residence provided at the datazone level (each datazone contains approximately 700 residents). We take all admissions for which COVID-19 was listed as primary or other contributing reason for admission, and aggregate to the datazone level before using the proportions of overlap between the datazones and the WWTP catchment shapefiles to aggregate to the geographical area of the each WWTP catchment.

## Variability

Our goal is to use data from WWTPs to observe changes in the growth rate of infections in the population. Since there are many factors that may affect the quantity of viral RNA in a sample, we cannot interpret every observed change in the reported value to be a true indicator of a change in the number of people infected. We often see the reported value fall to a fraction of what it was in the previous sample, only to jump to a high value again in the next sample. Rapid changes like this are too fast to be explained by transmission dynamics based on our current understanding of SARS-CoV-2 epidemiology.

Table 1 lists some of the causes of variability suggested in the literature. While each of these will have some effect, we generally lack sufficient data to control each of these factors; for the purpose of obtaining growth estimates it is more practical to accept that there is some inherent randomness in the wastewater sampling process. Our approach is to model the variability we expect to see at a given prevalence. We hypothesize that the variability between samples is predominantly the result of variability between the amount that infected individuals contribute to the wastewater sample, and this enables us to describe mathematically the distribution of outcomes we expect to see from a sample given for a given population prevalence. This distribution will provide a basis for modelling the underlying trend in prevalence.

Naively, it is natural to assume that the N1 gene copies contained in a 24-hour sample is contributed equally from every individual in the catchment who is infected and currently

**Table 1. Factors influencing temporal variation in the observed RNA count in wastewater samples.**

| Factor | Description |
|---|---|
| Movement | Prevalence may change on a daily or seasonal time scale depending on movements of infected people in or out of the catchment [19,32] |
| Temperature | May affect survival rate of RNA in the sewerage system [33] |
| Fluid flow | Household wastewater and rainfall that drains into sewerage pipes may cause variable amounts of dilution [19] |
| Industrial waste | Chemical waste can inhibit PCR assays [19] |
| qPCR variability | Outcomes of lab analysis can vary according to the equipment used and the precise details of how the test is performed [34] |
| Variant | Different variants induce different pathological responses which may alter shedding rates [27] |
| Vaccination | Vaccinated individuals may experience less severe outcomes from infection and therefore shed differently into the wastewater [35] |
| Demographic distribution of infections | Proportion of infections in older/younger population varies over time leading to variation in outcomes and shedding quantities [36] |

shedding the virus. In reality this may not be the case; firstly, there is some variability there in the amount each individual dispenses to wastewater (related to the amount of RNA fragments passing through the gastro-intestinal tract) [29–31], and secondly, as illustrated in Fig 1, fluids and fecal matter do not dilute perfectly throughout the sewerage influent, adding additional variability to the concentration of viral RNA received by the sampler. In an extreme case, the sampler may by chance pick up a large concentration of viral RNA from one highly infected individual giving an atypically large value [36].

Let us suppose that on a specific day there are $n$ individuals who are infected and contributing to some wastewater sample. We let $x_i$ represent the quantity of viral RNA contributed by individual $i$. We model the values $x_i$ for $i = 1, 2, ..., n$ as an i.i.d random variables from a distribution with mean $\mu$ and standard deviation $\sigma$. It follows from the central limit theorem that the cumulative quantity of viral RNA detected

$$v = \sum_{i=1}^{n} x_i, \tag{1}$$

is modelled as a random variable that follows a Normal distribution:

$$v \sim \mathcal{N}(n\mu, n\sigma^2). \tag{2}$$

If we assume that samples drawn from this distribution at any two times are independent, the variability over short time scales can be explained by a sufficiently high value of $n\sigma^2$.

This generalises to other data series including the number of reported cases or the number of hospital admissions. In both cases, each infected individual contributes either 0 or 1 to the

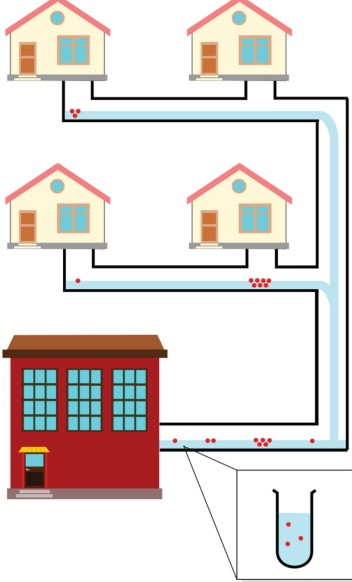

**Fig 1. Flow of viral RNA fragments.** RNA fragments shown as red circles. The amount from each household may vary across time and depend on severity of outcomes. When samples are taken, this variability is expected to affect the outcome of taking a sample. Composite sampling may reduce variability but the quantity of viral RNA detected may still be dominated by RNA originating from a single infection.

total count depending whether they report a positive test (for case count data) or have symptoms that lead to hospitalisation (for admissions data). The total count can therefore be modelled as an Binomial distribution and, for sufficiently large numbers contributing to the signal, approximated by the Normal distribution. The methods described in this paper are therefore applicable to a wide range of sources of epidemiological surveillance.

## Likelihood of underlying prevalence model

We allow $n$ to be a function of time, $n(t)$, representing the number of people who are shedding virus in wastewater on day $t$. Similarly $v(t)$ is the total quantity of viral RNA in the sampler on day $t$ (assuming a sample was taken). The function $f(t) = n(t)\mu$ transforms Eq 2 to

$$v(t) \sim \mathcal{N}(f(t), Df(t)). \tag{3}$$

where $D = \sigma^2/\mu$ is a property of the distribution of individual contributions, $x_i$, known as the *index of dispersion*. Note that we have assumed that this distribution is time-independent.

Our approach to estimating the epidemic trajectory from WWTP data is to find the function $f(t)$ and parameter $D$ that achieves the maximum likelihood for the given data. Representing the data as two vectors $\boldsymbol{t} = \{t_1, t_2, ..., t_k\}$, and $\boldsymbol{y} = \{y_1, y_2, ..., y_k\}$, where $t_i$ is the time the $i$th sample was taken and $y_i$ is the reported quantity of viral RNA, the best fitting parameter values are found by solving

$$\{\hat{f}, \hat{D}\} = \underset{f, D}{\operatorname{argmax}} \log \mathcal{L}(f, D \mid \boldsymbol{t}, \boldsymbol{y}), \tag{4}$$

where

$$\log \mathcal{L}(f, D \mid \boldsymbol{t}, \boldsymbol{y}) = -\frac{1}{2} \sum_{i=1}^{k} \log(2\pi Df(t_i)) + \frac{(y_i - f(t_i))^2}{Df(t_i)}. \tag{5}$$

The function $f(t)$ in Eq 4, can be any time series over the period where data were collected. Since our goal is to remove unwanted variability and reveal the underlying epidemic trajectory, we choose to constrain $f$ to a class of functions that are feasible given the epidemiology of the disease.

## Epidemic trajectory model

We limit our choice of the epidemic trajectory $f$ to the class of exponential functions with growth rates that change at discrete points in time. Supposing there are $m$ times when the exponential growth rate changes, hereafter referred to as change points, we let $\boldsymbol{c}$ be a vector of length $m$ representing the times of each change point, and $\boldsymbol{r}$ be a vector of length $m + 1$ where the $i$th entry is the growth during the interval from $c_i$ to the next change point. Thus we have

$$f(t; \boldsymbol{c}, \boldsymbol{r}, A_1) = \begin{cases} A_1 e^{r_1 t} & \text{if} \quad 0 \leq t < c_1 \\ A_2 e^{r_2 t} & \text{if} \quad c_1 \leq t < c_2 \\ \vdots & \\ A_{m+1} e^{r_{m+1} t} & \text{if} \quad c_m \leq t \end{cases} \tag{6}$$

where $A_1$ is a free parameter (the intercept), and the other $A_k$ are determined using the formula

$$A_k = \exp\left[\sum_{i=0}^{k}(r_i - r_{i-1})c_i\right] \tag{7}$$

to ensure that $f$ is continuous.

## Optimization procedure

An estimate for the epidemic trajectory and natural variability is found by solving Eq. 4 over the class of functions of the form expressed in Eq (6). The free parameters are the initial quantity of wastewater RNA ($A_1$), the number of times the growth rate changes ($m$), the times of each of these changes ($c$), the corresponding growth rates ($r$), and the index of dispersion ($D$). If we want to fit to data over a period containing several change points, the space of possible parameter combinations becomes very large and computationally expensive to optimize by brute force. We tried several off-the-shelf methods for solving the for optimization but these struggled to consistently produce useful output. We then developed our own heuristic procedure for solving Eq (4) which we detail here.

The outline of the procedure, illustrated in Fig 2, is as follows: A simple exponential curve is fitted to the first few data points using a hill-climbing algorithm. The remaining data points are added sequentially. Each time a new data point is added, the hill-climbing algorithm is re-applied to optimize the parameters. A candidate change point is considered between the most recent change point (or the start of the data if no change points have been added) and the time of the newly added data point. The hill-climbing algorithm is re-applied to optimize the parameters (including the time of the candidate change point and the growth rate after the change). If the inclusion of the change point yields an increase in the likelihood function, Eq. (5), that exceeds a given threshold value then the new change point is added to the parameter set of $f$. This is repeated until all the data have been added.

The choice of the threshold value has consequences for the outcome. If set close to zero it will produce a model that is highly sensitive to variability in the underlying data. Higher thresholds will find longer-term trends in the underlying data while being less sensitive to the

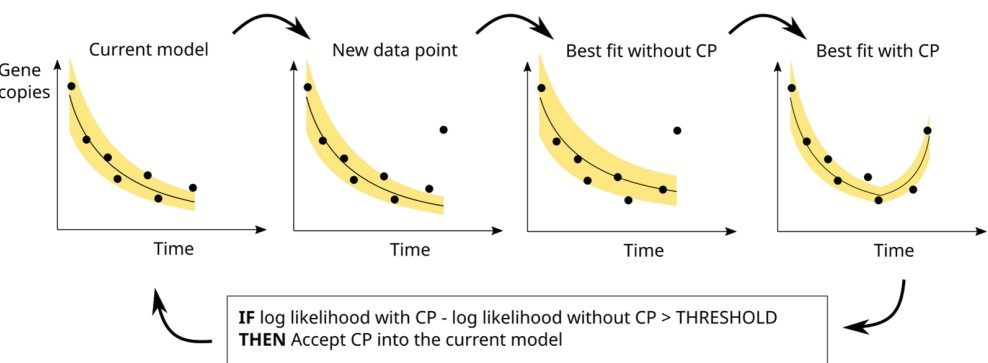

**Fig 2. Model fitting**. The model is fitted to the data by iteratively adding data points and optimizing the parameters using hill-climbing at each iteration. In each iteration a candidate change point (CP) is added. If there is an improvement in the likelihood resulting from the added exceeds some threshold value then the change point becomes part of the accepted model.

natural fluctuations of the wastewater signal. The choice of threshold value can only be determined when the goal of the analysis has been decided. In this paper the goal is to detect waves of SARS-CoV-2 infection as early as possible.

Here we describe the hill-climbing part of the algorithm. The first step selects a discrete set of possible values for each parameter. The dispersion parameter ($D$) can take values from 0.5% to 10% of the maximum quantity of viral RNA across the series at intervals of 0.5%, the time of each change point ($c$) can be any day between the midpoint of its current value and the previous change point (or start of the data), and the midpoint of its current value and the next change point (or end of the data at that iteration of the algorithm).

The initial quantity of wastewater RNA ($A_1$) exponential growth rates ($r$) are not included as parameters explicitly, instead we use the gene copy value at each change point and use them to calculate growth rates after the optimization. The range for the gene copy values is 100 equally spaced values from 0 to 1.5 multiplied by the maximum value in the series. The hill-climb proceeds by perturbing parameters one at a time to adjacent values within the ranges described. It then chooses to accept the perturbation which yielded the best improvement. This continues until no improvement can be made.

## Results

### Model fitting

We use the 10 largest WWTPS (by catchment population) to explore the effects of the threshold on the outcome. As shown in Fig 3, higher threshold values produce fewer changes in the modelled trajectory and correspondingly higher values for the dispersion (the measure of non-epidemic variability). The threshold value of 8 was provides outcomes that are neither too sensitive to natural variability yet still responsive to real changes in the epidemic trajectory. We applied the procedure with this threshold to 121 WWTPs, 1 was omitted for containing no positive values for the quantity of viral RNA. The quality of the outcome is measured by the mean log likelihood; the maximum likelihood result divided by the number of data points. This varied across sites, however, it did not vary by the number of samples in the data (Pearson's $r$,$p$ = 0.36) and there was no significant effect of the catchment area population size (Pearson's $r$,$p$ = 0.16), implying that the procedure can be applied successfully in geographical regions as small as 1500 people (S1 Fig).

### Real-time estimation of the epidemic growth rate

The real-time growth rate at time $t$ is the rate value ($r_{m+1}$, in the function $f$ given by Eq. (6)) obtained after fitting the model to time series up to and including time $t$. Fig 4A shows the estimated growth rates over time for one WWTP for three different threshold values. This example illustrates a general trend that we observe across all the treatment sites: that the responsiveness of the real-time estimate to changes in the data is dependent on the chosen threshold value.

As measure of the responsiveness to changes in the epidemic trajectory we count the number of periods of uninterrupted positive growth. We see that periods of growth tend to start earlier for lower thresholds, potentially providing an earlier warning that prevalence will rise, but also causing a larger frequency of "false alarms" where the direction of growth will change for a short period before changing again. From Fig 4 we see that the number of waves predicted is highly sensitive to the threshold value when it is below 6, while threshold values of 8 or greater provide relatively consistent outcomes. We therefore suggest that values from 6 to 8

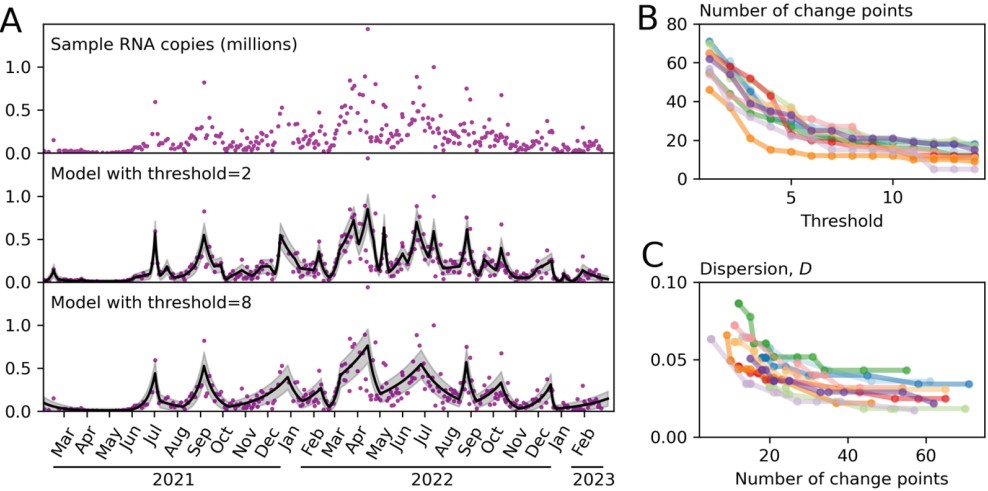

**Fig 3. Relationship between the sensitivity threshold and the number of trajectory changes.** (**A**) The top panel shows the raw data reporting the quantity of viral RNA in the collected samples. The panels below show the outcome of model fitting using maximum likelihood estimation for two values of the threshold hyperparameter. (**B**) The dependence of the number of change points on the threshold hyperparameter for the 10 largest WWTPs by catchment population. (**C**) Relationship between the number of change points and the index of dispersion. With fewer change points, variability around the mean must necessarily be larger in order to explain the distribution of observed quantities.

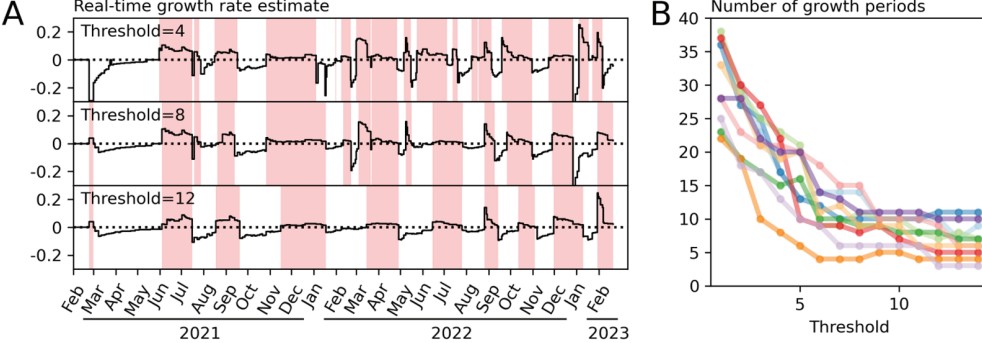

**Fig 4. Real-time growth rate estimates.** (**A**) Estimates for the Seafield WWTP at three threshold values. The solid line in these figures shows the estimated growth rate using the data available at each the given point in time. Higher thresholds are less responsive to high frequency variability in the data. Regions shaded red indicate periods uninterrupted positive growth. (**B**) The number of periods uninterrupted positive growth. Low threshold values produce a high frequency of "false alarms" - changes to the trajectory that are not sustained—higher threshold values are more stable but periods of positive growth tend to begin later.

provide high responsiveness while remaining insensitive to high frequency fluctuations in the data.

## Comparison to hospital admissions data

Throughout the Covid-19 epidemic the number of reported cases has provided a useful metric to help forecast hospital demand [37]. As the number of reported test results is highly dependent on test-seeking behaviour, wastewater can potentially be a less biased way to provide the same early warning. To address this we compare the growth rates obtained from

wastewater data to those obtained from hospital admission data to see if one time series predicts the other.

To obtain growth rates from hospital admission data we applied our method to time series data reporting the number of people admitted to hospital for COVID-19 who live in the catchment of each of the 10 largest WWTPs by catchment population (Fig 5A). We found that a threshold value of 8 provided a real-time growth rate that captured the major changes to the epidemic trajectory without being overly responsive to fluctuations in the data. It predicted a similar number of waves to the wastewater output for the same threshold value (within 1 across all WWTPs tested) and therefore provided outputs suitable for comparison. Fig 5B shows an example of this for one WWTP.

We calculated the correlation between the time series over a range of lags. A lag of length $l$ means that the growth rate value on day $t$ in the wastewater time series is paired with the growth rate value on day $t + l$ in the hospital admissions time series. The optimal lag is the value of $l$ that give the highest correlation between the two series. A positive optimal lag implies that the signal coming from wastewater data responds quicker to changes in the epidemic trajectory than the signal coming from admissions data. We find that the optimal lag is positive for 6 of the 10 sites. Taking the mean of the Pearson's $r$ values over the 10 sites we find that a lag of 2 days achieved the largest correlation (Fig 5C).

## Environmental factors

Rainfall entering the sewerage system is expected to dilute the concentration of viral material in the fluid leading to lower observed values. While this dilution effect will vary depending on the amount of rainfall, it can be difficult to observe directly as its magnitude is small in comparison to the variation in viral prevalence. To test whether fluid flow has an effect on the reported quantities of viral RNA, we compare fluid flow values to the $z$-scores of the observed viral RNA quantities with respect to the modelled distribution (using a sensitivity threshold value of 8). The $z$-score is calculated by subtracting the model mean from the observed

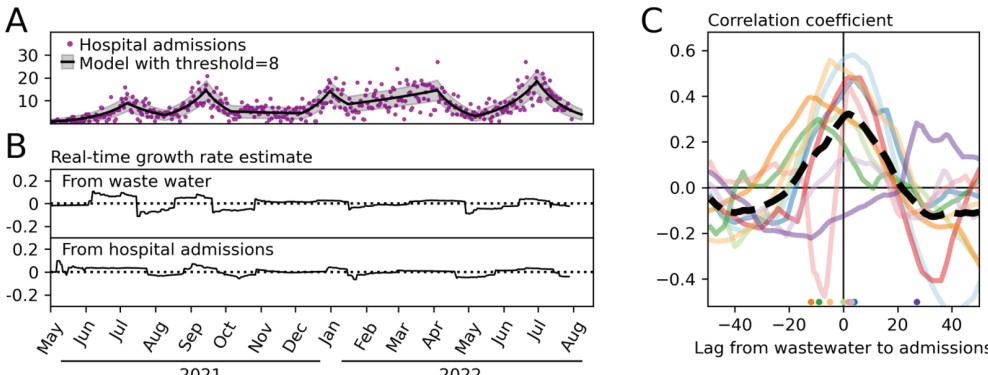

**Fig 5. Comparison to admissions data.** (**A**) Model fitted to data from hospital admissions of individuals living in the catchment of the Seafield WWTP using identical methods to those applied to wastewater data in previous figures. (**B**) The real time growth rate estimated using samples from the Seafield WWTP and admissions for the corresponding catchment area. (**C**) Pearson's $r$ correlation between the time-series produced in panel B and for the 10 largest WWTP catchments (by population) in Scotland over a range of lags. Circles are added at the bottom of the figure to show the value of the lag that yields the maximum correlation. The mean correlation over the 10 sites is shown as a dashed line. Positive lag implies that wastewater growth rates respond faster than those obtained from hospital admissions data.

value and dividing by the standard deviation, in principal removing the effects of underlying epidemic trajectory.

Of the 15 sites with more than 100 recorded flow values, correlation between fluid flow and higher than expected viral RNA was found in 5 (Pearson's $r<0$, $p<0.05$). In these cases the effect size is small in comparison to the overall amount of variability. For example, applying linear regression to data from Meadowhead, we find that that each additional 100 megalitres of flow per day reduces the mean sample value by 0.41 standard deviations (S2 Fig).

## Discussion

The goal of this work was to provide a methodology to interpret wastewater data and help inform policy decisions. A major challenge to decision makers is deciding when a change in the reported data indicates a genuine change in the underlying epidemic trajectory, particularly when the data are subject to high levels of natural variability. Our approach accounts for the variability between samples observed in the reported RT-qPCR result and uses it as the basis for a model fitting procedure. The main outcome is an estimate of the growth rate of the epidemic that updates each time new data become available. Applying this procedure to data from sampling locations across Scotland, we found that samples collected from wastewater detect changes to the epidemic trajectory approximately two days earlier than can be observed in hospital admissions.

Despite the recent growth of wastewater surveillance as a non-intrusive source of population health data, there are no clear guidelines around how these data should be used to inform policy. We note that deep analysis is not always needed; Polio, for example, is rare in the UK but confers a profound risk if it is not controlled - if the virus is detected then action should be taken [38]. For highly prevalent diseases, on the other hand, deciding when to take action is less straight-forward; there is always a risk of mistaking a random fluctuation for a real spike in incidence, and this may be a reason for some to distrust this valuable source of data. Our approach of estimating the growth rate of an underlying model resolves this problem.

We found on average that changes in the wastewater signal precede admissions by two days. Similar positive lags between wastewater and other indicators have been found previously [2–4,6,20], however, these results are based on retrospective analysis of time series data - they do not consider the accuracy of their predictions if they were to be deployed operationally in real time. Our methodology has been designed with the goal of operational use, that means providing information that can be acted upon in real time, potentially supporting resource management in hospitals (cancelling elective operations, increase staffing, stock up on oxygen). The 2-day lag found may provide more time, however, we warn that this is only an average; sometimes our method may produce false alarms, sometimes it may not detect change until after signal is seen in hospital admissions. Future work should explore how this method can be used in combination with other indicators.

While previous related literature has attempted to use environmental factors to explain signal variability [2,19,20], we have instead posited that most of the variability is caused by the non-dilution of particles containing large quantities of RNA. Future work could test this hypothesis experimentally. This could easily be done by installing two (or more) autosamplers at the same WWTP that are sampling at slightly different times (for example, offset by 30 minutes on a 1 hour sampling cycle) and comparing their outputs.

We highlight here some limitations of our analysis and areas for potential improvement. Firstly, the method has only been tested on SARS-CoV-2 data in Scotland during a period when prevalence was high. While the method can be applied in other settings we recommend

first validating the outputs against other sources of surveillance data in those settings. Secondly, we made a specific choice to explore the exponential function as the prevalence curve. This assumption is reasonable in most cases as infectious diseases dynamics are typically driven by branching processes parameterized by the effective reproductive number. There may, however, be situations where other functions should be considered for example if we are to take into account the duration of shedding [21,30]. Thirdly, adjusting the observed quantities of viral RNA using a normalisation factor (such as the fluid flow which we have shown to affect the model output) could potentially yield some improvement to the results [20,27,27,39, 40]. Similarly, adjusting the hospital admissions data prior analysis to remove known periodic patterns may yield an improvement.

One advantage of our method is its applicability to data from individual sites distributed over a range of geographical locations. Better spatial coverage achieves earlier warnings of the start of an epidemic wave and better targeting of control measures [41]. As wastewater surveillance grows, we may see samples being taken at more WWTPs or even within the sewerage network, reaching ever smaller populations [42,43]. While our results demonstrate that surveillance for Covid-19 can be useful over catchment populations as low as 1500, we don't know precisely where the limitations of the method lie. Future work should explore these limits, not just in terms of spatial resolution, but also the sampling frequency and underlying prevalence.

## Conclusion

We have contributed a novel procedure for estimating growth rates based on wastewater surveillance data and shown that it can provide an early warning for growing COVID-19 hospital admissions. We hope that this work contributes to the wider goal of improving the way we mitigate the risks of disease through wastewater surveillance. While we are confident that the methods developed here are applicable to a range of current settings, they exist to be improved upon and adapted to the specific diseases and challenges in wastewater monitoring. The ongoing challenge is to build on the work of many environmental and biological scientists that have made these data sources possible, to push the continued development of mathematical and statistical tools, and help support decision makers and the population by providing the best possible scientific analysis.

## Supporting information

**S1 Fig. Model fit does not appear to be affected by the number of samples or the population of the WWTP catchment.** The log likelihood of model using the maximal likelihood parameter values is divided by the number of data points to give the mean log likelihood. Each point in these plots represents a WWTP.
(TIFF)

**S2 Fig. The affect of fluid flow is small. We show the 10 largest WWTPs (by catchment population) for which flow data are available.** Each point in these plots represents a sample. The vertical axis shows the $z$-score for the sample with respect to the distribution that maximizes the model likelihood.
(TIFF)

## Acknowledgments

We are grateful to the Scottish Environmental Protection Agency, Scottish Water, and Public Health Scotland for making data for this study available. We thank our partners in the

Scottish Government Wastewater Monitoring Data Integration Subgroup for their insightful feedback.

## Author contributions

**Conceptualization:** Ewan Colman, Rowland Kao.

**Data curation:** Ewan Colman, Rowland Kao.

**Formal analysis:** Ewan Colman.

**Funding acquisition:** Rowland Kao.

**Investigation:** Ewan Colman.

**Methodology:** Ewan Colman, Rowland Kao.

**Project administration:** Ewan Colman, Rowland Kao.

**Resources:** Ewan Colman.

**Software:** Ewan Colman.

**Supervision:** Rowland Kao.

**Visualization:** Ewan Colman.

**Writing – original draft:** Ewan Colman.

**Writing – review & editing:** Rowland Kao.

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
