## [Decision Letter · Decision Letter 0]

7 Feb 2025

PONE-D-25-03721The impact of signal variability on epidemic growth rate estimation from wastewater surveillance dataPLOS ONE

Dear Dr. Colman,

Thank you for submitting your manuscript to PLOS ONE. After careful consideration, we feel that it has merit but does not fully meet PLOS ONE’s publication criteria as it currently stands. Therefore, we invite you to submit a revised version of the manuscript that addresses the points raised during the review process.

We look forward to receiving your revised manuscript.

Kind regards,

Muammar Qadafi

Academic Editor

PLOS ONE

Journal Requirements:

“The first author was supported by the Wellcome Trust and the National COVID-19 Wastewater Epidemiology Surveillance Programme.”

“We are grateful to the Scottish Environmental Protection Agency, Scottish Water, and Public Health Scotland for making data for this study available. We thank our partners in the Scottish Government Wastewater Monitoring Data Integration Subgroup for their insightful feedback. Financial support was received from The Wellcome Trust and the National COVID-19 Wastewater Epidemiology Surveillance Programme.

“The first author was supported by the Wellcome Trust and the National COVID-19 Wastewater Epidemiology Surveillance Programme.”

6. We note that you have indicated that there are restrictions to data sharing for this study. For studies involving human research participant data or other sensitive data, we encourage authors to share de-identified or anonymized data. However, when data cannot be publicly shared for ethical reasons, we allow authors to make their data sets available upon request. For information on unacceptable data access restrictions, please see http://journals.plos.org/plosone/s/data-availability#loc-unacceptable-data-access-restrictions.

7. We note that Figure 1 in your submission contain copyrighted images. All PLOS content is published under the Creative Commons Attribution License (CC BY 4.0), which means that the manuscript, images, and Supporting Information files will be freely available online, and any third party is permitted to access, download, copy, distribute, and use these materials in any way, even commercially, with proper attribution. For more information, see our copyright guidelines: http://journals.plos.org/plosone/s/licenses-and-copyright.

8. We notice that your supplementary figures are uploaded with the file type 'Figure'. Please amend the file type to 'Supporting Information'. Please ensure that each Supporting Information file has a legend listed in the manuscript after the references list.

Reviewers' comments:

Reviewer's Responses to Questions

**Comments to the Author**

1. Is the manuscript technically sound, and do the data support the conclusions?

Reviewer #1: No

Reviewer #2: Yes

2. Has the statistical analysis been performed appropriately and rigorously? 

Reviewer #1: No

Reviewer #2: Yes

3. Have the authors made all data underlying the findings in their manuscript fully available?

Reviewer #1: No

Reviewer #2: No

4. Is the manuscript presented in an intelligible fashion and written in standard English?

Reviewer #1: Yes

Reviewer #2: Yes

5. Review Comments to the Author

Reviewer #1: *Only SARS-CoV-2 N1 was detected. The topic should be "The impact of signal variability on epidemic growth rate estimation of SARS-CoV-2 N1 from wastewater surveillance data".

*Please follow the PLOS ONE's abstract format.

*The research gap is necessary for the last paragraph of the introduction.

*According to the methodology:

From February 2021 to the time of writing (February 2023), the Scottish Environment Protection Agency (SEPA) regularly took samples from wastewater treatment plants (WWTP) to detect fragments of SARS-Cov-2 virus RNA. Samples from sewerage influent were taken at 122 sampling locations across Scotland, approximately two to three days each week, depending on the location. *The study setting's population was wastewater from Scotland's WWTP. However, the sampling technique is crucial to ensure that the samples are appropriate for a representative of Scotland; please justify the samples from WWTP (the population) and the purposive selection or random selection for 122 sampling locations.

If the data cannot be published, please add the statement, "Hospitalisation data were provided to us under a strict data sharing policy and can not be made public". But sampling technique is still necessary.

*10 largest WWTPS shown in the result but not in the methodology, please add the sampling technique for 10 largest WWTPS.

*Please justify for data analysis details, such as the Likelihood of the underlying prevalence model and Epidemic trajectory model or other analysis.

*Please avoid using we and our, but use researchers and researchers instead.

*The conclusion could be more comprehensive based on how a strategic planner could improve the outcome or policies based on the study's results.

*It is suggested that limitations and recommendations for further studies be added.

Reviewer #2: Before a manuscript can be published major issues must be addressed by the author.

1. Abstract writing is not in accordance with the standard.

2. There are many similar studies, the author is asked to write a gap review to clarify the novelty of the research.

3. State of the art is not detailed enough, introduction is too short.

4. The discussion is not deep enough and without enough comparisons.

5. The discussion presented is not only about modeling but the author must also discuss the environmental conditions/characteristics of wastewater and its relationship with epidemic growth.

6. Conclusions must be presented with numbers / research results.

6. PLOS authors have the option to publish the peer review history of their article (what does this mean?). If published, this will include your full peer review and any attached files.

Reviewer #1: No

Reviewer #2: No

---

## [Author Response · Author response to Decision Letter 1]

28 Feb 2025

We would like to thank the editor and reviewers for their carful reading of our manuscript, and for their insightful comments and suggestions. Below is our response to each comment.

Sincerely,

Ewan Colman

~~~~~~~~~~~

Editor

We have reviewed the formatting guidelines and made the required changes

Code is available at https://github.com/EwanColman/Estimating-epidemic-growth-rate-from-wastwater-data

“The first author was supported by the Wellcome Trust and the National COVID-19 Wastewater Epidemiology Surveillance Programme.”

The amended funding statement is as follows:

EC was supported by Wellcome Trust (grant no. 209818/Z/17/Z) and the National COVID-19 Wastewater Epidemiology Surveillance Programme. RK received no specific funding for this work. The funders had no role in study design, data collection and analysis, decision to publish, or preparation of the manuscript. There was no additional external funding received for this study.

“We are grateful to the Scottish Environmental Protection Agency, Scottish Water, and Public Health Scotland for making data for this study available. We thank our partners in the Scottish Government Wastewater Monitoring Data Integration Subgroup for their insightful feedback. Financial support was received from The Wellcome Trust and the National COVID-19 Wastewater Epidemiology Surveillance Programme.

“The first author was supported by the Wellcome Trust and the National COVID-19 Wastewater Epidemiology Surveillance Programme.”

Please see the amended statement above. We have removed funders from the acknowledgements.

6. We note that you have indicated that there are restrictions to data sharing for this study. For studies involving human research participant data or other sensitive data, we encourage authors to share de-identified or anonymized data. However, when data cannot be publicly shared for ethical reasons, we allow authors to make their data sets available upon request. For information on unacceptable data access restrictions, please see http://journals.plos.org/plosone/s/data-availability#loc-unacceptable-data-access-restrictions.

Hospital admissions data are now available in a de-identified anonymized form at https://doi.org/10.6084/m9.figshare.28512479.v1

7. We note that Figure 1 in your submission contain copyrighted images. All PLOS content is published under the Creative Commons Attribution License (CC BY 4.0), which means that the manuscript, images, and Supporting Information files will be freely available online, and any third party is permitted to access, download, copy, distribute, and use these materials in any way, even commercially, with proper attribution. For more information, see our copyright guidelines: http://journals.plos.org/plosone/s/licenses-and-copyright.

Figure 1 is an original image created by the authors.

8. We notice that your supplementary figures are uploaded with the file type 'Figure'. Please amend the file type to 'Supporting Information'. Please ensure that each Supporting Information file has a legend listed in the manuscript after the references list.

We have now uploaded the supplementary figures as 'Supporting Information'

Reviewer 1

*Only SARS-CoV-2 N1 was detected. The topic should be "The impact of signal variability on epidemic growth rate estimation of SARS-CoV-2 N1 from wastewater surveillance data".

Thank you for this suggestion. Our goal is to estimate the growth rate of COVID-19. The SARS-CoV-2 N1 gene count is the data we use to inform that estimate (not the thing we are aiming to estimate). We have changed the title to "The impact of signal variability on COVID-19 epidemic growth rate estimation from wastewater surveillance data"

*Please follow the PLOS ONE's abstract format.

Could the reviewer please be more specific in this request?

The abstract already meets the requirements for PLoS One: (a) Describe the main objectives of the study, (b) Explain how the study was done, and (c) Summarize the most important results and their significance. Moreover, it follows guidelines published by PLoS on how to write a good abstract https://plos.org/resource/how-to-write-a-great-abstract/

*The research gap is necessary for the last paragraph of the introduction.

We have added the following statement addressing the research gap. "Within the existing literature, there are no studies that address the problem of inherent signal variability in designing a wastewater-based early warning system."

*According to the methodology:

From February 2021 to the time of writing (February 2023), the Scottish Environment Protection Agency (SEPA) regularly took samples from wastewater treatment plants (WWTP) to detect fragments of SARS-Cov-2 virus RNA. Samples from sewerage influent were taken at 122 sampling locations across Scotland, approximately two to three days each week, depending on the location. *The study setting's population was wastewater from Scotland's WWTP. However, the sampling technique is crucial to ensure that the samples are appropriate for a representative of Scotland; please justify the samples from WWTP (the population) and the purposive selection or random selection for 122 sampling locations.

We do not claim that the data used is representative of any wider population. We only suggest that the data coming from a single WWTP is in some way representative of the population of its catchment area. However, to give some clarification we have added the following to the data section of the methods:

"Combining the catchment areas of all 122 WWTPs covers the households of approximately 4500000 people, 82\% of the population of Scotland."

If the data cannot be published, please add the statement, "Hospitalisation data were provided to us under a strict data sharing policy and can not be made public". But sampling technique is still necessary.

This is already included in the data availability statement. Regarding sampling, in the data section of the methods we have changed "The hospital admission database includes information on individuals admitted to hospital in Scotland" to "The hospital admission database includes information on all individuals admitted to hospital in Scotland"

*10 largest WWTPS shown in the result but not in the methodology, please add the sampling technique for 10 largest WWTPS.

We have added "by catchment population" each time we refer to the largest WWTP. Additionally, we have added to the data section of the methods "The ten most populated catchments cover 47% [of the population]."

*Please justify for data analysis details, such as the Likelihood of the underlying prevalence model and Epidemic trajectory model or other analysis.

With apologies to the reviewer, I do not understand what is being requested here. Please could you specify the part of the article to which it refers, and why it does not meet the criteria for publication?

*Please avoid using we and our, but use researchers and researchers instead.

My understanding is that it is not within the remit of a PLoS one reviewer to dictate the written style the article. We (the researchers) think that articles written in the first person are easier to follow.

*The conclusion could be more comprehensive based on how a strategic planner could improve the outcome or policies based on the study's results.

We address this with new material added to the discussion:

We found on average that changes in the wastewater signal precede admissions by two days. Similar positive lags between wastewater and other indicators have been found previously, however, these results are based on retrospective analysis of time series data - they do not consider the accuracy of their predictions if they were to be deployed operationally in real time. Our methodology has been designed with the goal of operational use, that means providing information that can be acted upon in real time, potentially supporting resource management in hospitals (cancelling elective operations, increase staffing, stock up on oxygen). The 2-day lag found may provide more time, however, we warn that this is only an average; sometimes our method may produce false alarms, sometimes it may not detect change until after signal is seen in hospital admissions. Future work should explore how this method can be used in combination with other indicators.

*It is suggested that limitations and recommendations for further studies be added.

This is addressed in the paragraph above, and in another new paragraph added to the discussion:

While previous related literature has attempted to use environmental factors to explain signal variability, we have instead posited that most of the variability is caused by the non-dilution of particles containing large quantities of RNA. Future work could test this hypothesis experimentally. This could easily be done by installing two (or more) autosamplers at the same WWTP that are sampling at slightly different times (for example, offset by 30 minutes on a 1 hour sampling cycle) and comparing their outputs.

Reviewer 2

Before a manuscript can be published major issues must be addressed by the author.

1. Abstract writing is not in accordance with the standard.

The abstract already meets the requirements for a PLoS One abstract: (a) Describe the main objectives of the study, (b) Explain how the study was done, and (c) Summarize the most important results and their significance. Moreover, it follows guidelines published by PLoS on how to write a good abstract https://plos.org/resource/how-to-write-a-great-abstract/

2. There are many similar studies, the author is asked to write a gap review to clarify the novelty of the research.

We have added the following statement addressing the research gap. "Within the existing literature, there are no studies that address the problem of inherent signal variability in designing a wastewater-based early warning system."

3. State of the art is not detailed enough, introduction is too short.

Could the reviewer please specify what details are required that are not included? It was our intention to keep the introduction short, only including necessary information while directing the reader to relevant literature for further details. If the reviewer would like to suggest other published papers that have similar aims to ours then we would be happy to include them.

4. The discussion is not deep enough and without enough comparisons.

We address this with new material added to the discussion:

We found on average that changes in the wastewater signal precede admissions by two days. Similar positive lags between wastewater and other indicators have been found previously, however, these results are based on retrospective analysis of time series data - they do not consider the accuracy of their predictions if they were to be deployed operationally in real time. Our methodology has been designed with the goal of operational use, that means providing information that can be acted upon in real time, potentially supporting resource management in hospitals (cancelling elective operations, increase staffing, stock up on oxygen). The 2-day lag found may provide more time, however, we warn that this is only an average; sometimes our method may produce false alarms, sometimes it may not detect change until after signal is seen in hospital admissions. Future work should explore how this method can be used in combination with other indicators.

5. The discussion presented is not only about modeling but the author must also discuss the environmental conditions/characteristics of wastewater and its relationship with epidemic growth.

We have added the following paragraph to the discussion to address this:

While previous related literature has attempted to use environmental factors to explain signal variability, we have instead posited that most of the variability is caused by the non-dilution of particles containing large quantities of RNA. Future work could test this hypothesis experimentally. This could easily be done by installing two (or more) autosamplers at the same WWTP that are sampling at slightly different times (for example, offset by 30 minutes on a 1 hour sampling cycle) and comparing their outputs.

6. Conclusions must be presented with numbers / research results.

The conclusion now starts with the following:

We have contributed a novel procedure for estimating growth rates based on wastewater surveillance data and shown that it can provide an early warning for growing COVID-19 hospital admissions.

---

## [Decision Letter · Decision Letter 1]

4 Mar 2025

PONE-D-25-03721R1The impact of signal variability on COVID-19 epidemic growth rate estimation from wastewater surveillance dataPLOS ONE

Dear Dr. Colman,

Thank you for submitting your manuscript to PLOS ONE. After careful consideration, we feel that it has merit but does not fully meet PLOS ONE’s publication criteria as it currently stands. Therefore, we invite you to submit a revised version of the manuscript that addresses the points raised during the review process.

We look forward to receiving your revised manuscript.

Kind regards,

Muammar Qadafi

Academic Editor

PLOS ONE

Journal Requirements:

Reviewers' comments:

Reviewer's Responses to Questions

**Comments to the Author**

1. If the authors have adequately addressed your comments raised in a previous round of review and you feel that this manuscript is now acceptable for publication, you may indicate that here to bypass the “Comments to the Author” section, enter your conflict of interest statement in the “Confidential to Editor” section, and submit your "Accept" recommendation.

Reviewer #1: (No Response)

2. Is the manuscript technically sound, and do the data support the conclusions?

Reviewer #1: Partly

3. Has the statistical analysis been performed appropriately and rigorously? 

Reviewer #1: N/A

4. Have the authors made all data underlying the findings in their manuscript fully available?

Reviewer #1: No

5. Is the manuscript presented in an intelligible fashion and written in standard English?

Reviewer #1: Yes

6. Review Comments to the Author

Reviewer #1: Reviewer 1's comments

Comment 1: *Only SARS-CoV-2 N1 was detected. The topic should be "The impact of signal variability on epidemic growth rate estimation of SARS-CoV-2 N1 from wastewater surveillance data".

Researchers' Response 1: Thank you for this suggestion. Our goal is to estimate the growth rate of COVID-19. The SARS-CoV-2 N1 gene count is the data we use to inform that estimate (not the thing we are aiming to estimate). We have changed the title to "The impact of signal variability on COVID-19 epidemic growth rate estimation from wastewater surveillance data"

Comment 1 for the revision version: Acceptable, and the revised version is better than the original version.

(the previous topic "The impact of signal variability on epidemic growth rate estimation from wastewater surveillance data".)

Comment 2: *Please follow PLOS ONE's abstract format. Could the reviewer please be more specific in this request?

Researchers' Response 2: The abstract already meets the requirements for PLoS One: (a) Describe the main objectives of the study, (b) Explain how the study was done, and (c) Summarize the most important results and their significance. Moreover, it follows guidelines published by PLoS on how to write a good abstract https://plos.org/resource/how-to-write-a-greatabstract/

Comment 2 for Revision 1: Please add the abstract sections for the sub-headings of the introduction (background), methods, results, and conclusion. Data collection (sampling technique, such as purposive, convenience, or random sampling) and analysis (such as RT-qPCR) are necessary.

Comment 3: *The research gap is necessary for the last paragraph of the introduction.

Researchers' Response 3: We have added the following statement addressing the research gap. "Within the existing literature, there are no studies that address the problem of inherent signal variability in designing a wastewater-based early warning system."

Comment 3 for Revision Version: Acceptable, and the revised version is better than the original version.

Comment 4: *According to the methodology: From February 2021 to the time of writing (February 2023), the Scottish Environment Protection Agency (SEPA) regularly took samples from wastewater treatment plants (WWTP) to detect fragments of SARS-Cov-2 virus RNA. Samples from sewerage influent were taken at 122 sampling locations across Scotland, approximately two to three days each week, depending on the location.

Researchers' Response 4: We do not claim that the data used is representative of any wider population. We only suggest that the data coming from a single WWTP is in some way representative of the population of its catchment area. However, to give some clarification we have added the following to the data section of the methods: "Combining the catchment areas of all 122 WWTPs covers the households of approximately 4500000 people, 82% of the population of Scotland."

Comment 4 for Revision Version:: Acceptable, and the revised version is better than the original version.

Comment 5: If the data cannot be published, please add the statement, "Hospitalisation data were provided to us under a strict data sharing policy and can not be made public". But sampling technique is still necessary.

Researchers' Response 5: This is already included in the data availability statement. Regarding sampling, in the data section of the methods we have changed "The hospital admission database includes information on individuals admitted to hospital in Scotland" to "The hospital admission database includes information on all individuals admitted to hospital in Scotland"

Comment 5: Acceptable, and the revised version is better than the original version.

Comment 6: *10 largest WWTPS shown in the result but not in the methodology, please add the sampling technique for 10 largest WWTPS. We have added "by catchment population" each time we refer to the largest WWTP. Additionally, we have added to the data section of the methods "The ten most populated catchments cover 47% [of the population]." *Please justify for data analysis details, such as the Likelihood of the underlying prevalence model and Epidemic trajectory model or other analysis.

Researchers' Response 6: With apologies to the reviewer, I do not understand what is being requested here. Please could you specify the part of the article to which it refers, and why it does not meet the criteria for publication?

Comment for Revision: Please add sampling techniques, such as random sampling, purposive sampling, or convenience sampling. Also, the data analysis using RT-qPCR. Please do not forget to adding the words, data collection and data analysis.

Comment 7: *Please avoid using we and our, but use researchers and researchers instead.

Researchers' Response 7: My understanding is that it is not within the remit of a PLoS one reviewer to dictate the written style the article. We (the researchers) think that articles written in the first person are easier to follow.

Comment 7 for Revision Version: No Comments.

Comment 8: *The conclusion could be more comprehensive based on how a strategic planner could improve the outcome or policies based on the study's results.

Researchers' Response 8: We address this with new material added to the discussion: We found on average that changes in the wastewater signal precede admissions by two days. Similar positive lags between wastewater and other indicators have been found previously, however, these results are based on retrospective analysis of time series data - they do not consider the accuracy of their predictions if they were to be deployed operationally in real time. Our methodology has been designed with the goal of operational use, that means providing information that can be acted upon in real time, potentially supporting resource management in hospitals (cancelling elective operations, increase staffing, stock up on oxygen). The 2-day lag found may provide more time, however, we warn that this is only an average; sometimes our method may produce false alarms, sometimes it may not detect change until after signal is seen in hospital admissions. Future work should explore how this method can be used in combination with other indicators.

Comment 8 for the Revision Version: Acceptable, and the revised version is better than the original version.

Comment 9: *It is suggested that limitations and recommendations for further studies be added.

Researchers' Response 9: This is addressed in the paragraph above, and in another new paragraph added to the discussion: While previous related literature has attempted to use environmental factors to explain signal variability, we have instead posited that most of the variability is caused by the non-dilution of particles containing large quantities of RNA. Future work could test this hypothesis experimentally. This could easily be done by installing two (or more) autosamplers at the same WWTP that are sampling at slightly different times (for example, offset by 30 minutes on a 1 hour sampling cycle) and comparing their outputs.

Comment 9: Acceptable, and the revised version is better than the original version.

7. PLOS authors have the option to publish the peer review history of their article (what does this mean?). If published, this will include your full peer review and any attached files.

Reviewer #1: No

---

## [Author Response · Author response to Decision Letter 2]

13 Mar 2025

PLOS One editorial office,

We would like to thank the editor and reviewer for their careful reading of our manuscript, and for their comments and suggestions. Below is our response to each comment.

Sincerely,

Ewan Colman

~~~~~~~~~~~

Editor and reviewer comments are written as indented text.

Editor

We checked all references, identified one that needed correcting, and corrected it.

Reviewer 1

4. Have the authors made all data underlying the findings in their manuscript fully available?

Reviewer #1: No

We have made data available in an anonymised de-identified format. The data statement we have provided is as follows

RNA count from wastewater data are available from the Scottish Environmental Protection Agency, https://informatics.sepa.org.uk/RNAmonitoring/

Hospital admissions data in de-identified format are available from https://doi.org/10.6084/m9.figshare.28512479.v1

Treatment site catchment data are available in the code repository, https://github.com/EwanColman/Estimating-epidemic-growth-rate-from-wastwater-data

*Comment 2 for Revision 1: Please add the abstract sections for the sub-headings of the introduction (background), methods, results, and conclusion. Data collection (sampling technique, such as purposive, convenience, or random sampling) and analysis (such as RT-qPCR) are necessary.

I do not think this is a requirement of PLOS one submissions. In fact, most PLOS one articles that I have seen do not follow this format. Perhaps the reviewer could provide a constructive reason why they think this format would improve the manuscript?

Similarly, the reviewer is asking for the method of measurement to be mentioned (we already say what is measured “the number of copies of the N1”). I do not agree with this change. Could they please clarify how this would improve the abstract?

To better describe the data used in this study we have changed “The real-time growth rate of the SARS-CoV-2 prevalence is estimated at 121 wastewater sampling sites” to “The real-time growth rate of the SARS-CoV-2 prevalence is estimated at all 121 wastewater sampling sites”

*Comment for Revision: Please add sampling techniques, such as random sampling, purposive sampling, or convenience sampling. Also, the data analysis using RT-qPCR. Please do not forget to adding the words, data collection and data analysis.

We addressed this comment in the previous response. We clarified that we had analysed data from all the available wastewater treatment sites. I would very much like to understand what remains unclear to the reviewer after reading our methods and the cited references, particularly ref. 25, and extensive description of the dataset used.

While you could call this a convenience sample, talking in that terminology may confuse readers as it implies that a statistic will be calculated on that sample and then extrapolated to a broader population. This does not describe the analysis we have performed.

---

## [Editor Report · Decision Letter 2]

16 Mar 2025

The impact of signal variability on COVID-19 epidemic growth rate estimation from wastewater surveillance data

PONE-D-25-03721R2

Dear Dr. Colman,

We’re pleased to inform you that your manuscript has been judged scientifically suitable for publication and will be formally accepted for publication once it meets all outstanding technical requirements.

Kind regards,

Muammar Qadafi

Academic Editor

PLOS ONE
---

## [Editor Report · Acceptance letter]

PONE-D-25-03721R2

PLOS ONE

Dear Dr. Colman,

I'm pleased to inform you that your manuscript has been deemed suitable for publication in PLOS ONE. Congratulations! Your manuscript is now being handed over to our production team.

Kind regards,

on behalf of

Dr. Muammar Qadafi

Academic Editor

PLOS ONE